# Assessing the global risk of typhoid outbreaks caused by extensively drug resistant Salmonella Typhi

Joseph Walker [1] ✉, Chrispin Chaguza[1,2], Nathan D. Grubaugh [1,2,3], Megan Carey [4,5], Stephen Baker[4,6], Kamran Khan[7,8], Isaac I. Bogoch[7,9] & Virginia E. Pitzer [1,2]

Since its emergence in 2016, extensively drug resistant (XDR) *Salmonella enterica* serovar Typhi (*S.* Typhi) has become the dominant cause of typhoid fever in Pakistan. The establishment of sustained XDR *S.* Typhi transmission in other countries represents a major public health threat. We show that the annual volume of air travel from Pakistan strongly discriminates between countries that have and have not imported XDR *S.* Typhi in the past, and identify a significant association between air travel volume and the rate of between-country movement of the H58 haplotype of *S.* Typhi from fitted phylogeographic models. Applying these insights, we analyze flight itinerary data cross-referenced with model-based estimates of typhoid fever incidence to identify the countries at highest risk of importation and sustained onward transmission of XDR *S.* Typhi. Future outbreaks of XDR typhoid are most likely to occur in countries that can support efficient local *S.* Typhi transmission and have strong travel links to regions with ongoing XDR typhoid outbreaks (currently Pakistan). Public health activities to track and mitigate the spread of XDR *S.* Typhi should be prioritized in these countries.

Typhoid fever, a human disease caused by *Salmonella enterica* serovar Typhi (*S.* Typhi), is estimated to cause tens of millions of cases and over 100 thousand deaths in low- and middle-income countries (LMICs) each year[1,2]. *S.* Typhi is primarily transmitted through the fecal-oral route (usually contaminated food and water) and is rarely found in settings with widespread access to clean water, sanitation, and hygiene infrastructure. Consequently, the burden of typhoid fever is concentrated in Africa and South and Southeast Asia[1–5]. Clinically, cases of typhoid fever are marked by bacteremia and an array of symptoms that may be difficult to distinguish from other febrile diseases, with the potential for gastrointestinal perforation, sepsis, and death in severe cases[6].

The development of effective antimicrobial regimens in the twentieth century transformed typhoid treatment, although the emergence and evolution of antimicrobial resistance (AMR) has repeatedly undermined these gains. Chloramphenicol quickly became the standard treatment for typhoid fever after reports of efficacy were published in 1948, but reports of treatment failure quickly emerged, and in the 1970s outbreaks of chloramphenicol-resistant typhoid swept India, Vietnam, Peru, and Korea[7–10]. By this time, ampicillin and trimethoprim-sulfamethoxazole had joined chloramphenicol in the arsenal of first-line treatments, but resistance to all three drugs, known as multi-drug resistance (MDR), was first detected in a large 1972 outbreak in Mexico[11].

[1]Department of Epidemiology of Microbial Diseases, Yale School of Public Health, New Haven, CT, USA. [2]Yale Institute for Global Health, Yale University, New Haven, CT, USA. [3]Department of Ecology and Evolutionary Biology, Yale University, New Haven, CT, USA. [4]Cambridge Institute of Therapeutic Immunology and Infectious Disease (CITIID), University of Cambridge School of Clinical Medicine, Cambridge, UK. [5]Department of Infection Biology, Faculty of Infectious and Tropical Diseases, London School of Hygiene & Tropical Medicine, London, UK. [6]Human Immunology Laboratory, International AIDS Vaccine Initiative, London, UK. [7]Department of Medicine, University of Toronto, Toronto, ON, Canada. [8]BlueDot, Toronto, ON, Canada. [9]Divisions of Infectious Diseases and General Internal Medicine, Toronto General Hospital, University Health Network, Toronto, ON, Canada. ✉e-mail: jo.walker@yale.edu

The MDR-associated H58 haplotype of *S*. Typhi emerged in India[12,13] in the 1980s, and now causes the majority of new typhoid fever cases in South Asia, East Africa, and parts of Southeast Asia, where its resistance to the most commonly used antimicrobials, and a potential growth advantage even in the absence of antimicrobials, allowed it to displace previously circulating genotypes[12–18]. After fluoroquinolones became the recommended treatment option for typhoid in the 1990s, fluoroquinolone nonsusceptibility (FQNS) emerged and is now ubiquitous across South Asia and rapidly increasing in parts of Africa and Southeast Asia[11,12,19]. In India, Nepal, and Bangladesh, the expansion of FQNS has coincided with a gradual decline in the frequency of MDR cases due to a shift in selective pressure[12,20].

Since 2016, Pakistan has experienced an outbreak of typhoid caused by extensively drug resistant (XDR) *S*. Typhi, after an MDR H58 strain acquired resistance to fluoroquinolones and third generation cephalosporins through an IncY plasmid[21–23]. The only remaining effective treatments for XDR *S*. Typhi are azithromycin, a widely used oral antimicrobial with broad activity, and carbapenems, which must be administered intravenously in better-equipped clinical settings, imposing a significant burden on patients and health systems. Effectively untreatable typhoid is a future possibility: azithromycin-resistant typhoid has independently emerged in multiple countries[24,25], and resistance to carbapenems has been observed in non-typhoidal *Salmonella*[26]. In at least six instances, the genes conferring fluoroquinolone and cephalosporin resistance have been integrated from the IncY plasmid into the *S*. Typhi chromosome itself, eliminating the fitness costs of plasmid carriage and making it more likely that resistance will be maintained even in the absence of continued exposure to these antimicrobials[27].

The geographic scope of the typhoid outbreak caused by XDR *S*. Typhi has expanded since its initial detection in Hyderabad, Pakistan. Cases of XDR *S*. Typhi have been detected in most Pakistani provinces, and are now the primary cause of typhoid fever nationwide[22,28]. Cases of XDR *S*. Typhi following travel from Pakistan have been diagnosed elsewhere in the Middle East[29–31] as well as in North America[32,33], Europe[21,34–36], Australia[37,38], and Taiwan[39]. While XDR *S*. Typhi has not been diagnosed locally in India, the detection of a case in the United Kingdom (UK) following travel from India implies at least one introduction event from Pakistan in the past, and raises the possibility of cryptic local transmission[27]. Between November 2019 and October 2020, 9 XDR typhoid cases among patients with no history of recent international travel were diagnosed in the United States (US), with genomic sequencing confirming a match between these isolates and the 4.3.1.1.P1 genotype (a subclade of H58 genotype 4.3.1) circulating in Pakistan[32]. In February 2022, the Chinese Center for Disease Control identified an outbreak of 23 XDR *S*. Typhi cases linked to contaminated water in a suburban Beijing apartment building, with genomic sequencing again implicating the 4.3.1.1.P1 genotype[40]. In the same month, a case of XDR typhoid caused by genotype 4.3.1.1.P1 was diagnosed in Hong Kong; it is unclear if this case, which did not report any recent travel outside of Hong Kong, has any connection with the Beijing outbreak[41].

In this analysis, our primary objective was to identify countries at the greatest risk for experiencing outbreaks and sustained spread of XDR typhoid, given the flow of air travel from Pakistan and the efficiency of local *S*. Typhi transmission. As a secondary objective, we sought to evaluate the historic role of air travel in the global spread of novel typhoid lineages by fitting phylogeographic models to whole-genome sequencing data of the H58 typhoid haplotype.

## Results

### Air travel patterns predict the global spread of AMR *S*. Typhi
There are 16 countries that are known to have imported XDR *S*. Typhi since it emerged in Pakistan in 2016 (Fig. 1 and Supplementary Table 1). To evaluate the relationship between air travel patterns and XDR

typhoid importation, we used data from the International Air Travel Association (IATA) to calculate the total number of air travelers that flew from Pakistan to 200 countries in 2019. We found that countries with known XDR *S*. Typhi importations received a median of 84,507 (IQR: 24,805–221,937) air travelers from Pakistan in 2019, while the other 185 countries received a median of only 252 annual passengers (IQR: 25–2996). Of the ten most popular country destinations for international flights from Pakistan, seven are known to have imported at least one case of XDR *S*. Typhi. Case ascertainment, investigation, and reporting may be more consistent in high-income settings, which received 87% of international air travelers from Pakistan in 2019 (Fig. 2) and include 14 of the 16 countries known to have imported XDR *S*. Typhi (Supplementary Table 1), due to greater medical and laboratory infrastructure and the unusualness of enteric fever cases in these settings. When we only consider these high-income countries, our finding of higher travel volume from Pakistan to countries with known XDR *S*. Typhi importations persist (Supplementary Fig. 1).

To further investigate the factors that drive the international spread of novel *S*. Typhi lineages, we used BEAST to fit discrete Bayesian phylogeographic models to 932 H58 *S*. Typhi genomes sampled from 21 countries between 1983 and 2013. Full details of our approach, including genome processing and alignment, subsampling, selection of parameters and prior distributions, and pooling of estimates can be found in the "Methods" section below. We used a generalized linear model (GLM) approach[42] within the BEAST environment to parameterize lineage transition rates between country pairs, using modeled estimates of air-travel volume in 2010[43], and an indicator variable for the presence of a shared land border as candidate predictors. To minimize bias, we performed three independent BEAST runs for five random subsamples of genomes from each country, for a total of 15 analyses. The maximum clade credibility tree and phylogeographic movements from one of the five subsets of genomes is shown in Fig. 3.

Across the 15 pooled BEAST runs, air travel volume (after log-transformation and standardization) had a posterior probability of inclusion in the model of 99.92% (Bayes factor = 3015.35, the ratio of posterior and prior odds of inclusion), and a median regression coefficient of 1.314 (95% posterior credible interval [pCI]: 0.636–2.042). In this context, a positive coefficient indicates a higher estimated rate of lineage transfer between country pairs with greater air passenger flows, while the high posterior inclusion probability and Bayes factor indicate strong statistical support for an association between air travel and lineage movement. The presence of a shared land border between countries, with a posterior inclusion probability of 79.86% (Bayes factor = 9.57) and a median regression coefficient of 0.851 (95% pCI: −2.07–2.55), was a less reliable predictor of H58 lineage movement between countries. Estimates from individual genome subsamples were comparable to each other (Supplementary Fig. 2).

The parallel findings above, obtained using distinct approaches and data sources, indicate that the volume of incoming air travel from Pakistan should be a reliable proxy for countries' risk of importing cases of XDR *S*. Typhi in a given period of time. This validates our approach of using air travel to predict the risk of future XDR *S*. Typhi outbreaks and sustained transmission.

### Countries at elevated risk of XDR typhoid outbreaks
Of the ten countries that received the most air travelers from Pakistan in 2019, which collectively account for 83% of the country's outgoing travel volume, four are located in the Middle East & North Africa, and two countries each are in the East Asia & Pacific, North America, and Europe & Central Asia regions (Supplementary Table 2). Of these ten countries, only Saudi Arabia, Turkey, and Malaysia have not reported a case of XDR typhoid. Since only a fraction of typhoid fever cases seek medical care and are accurately tested, the possibility of additional undetected importations cannot be ruled out[44]. If the XDR *S*. Typhi

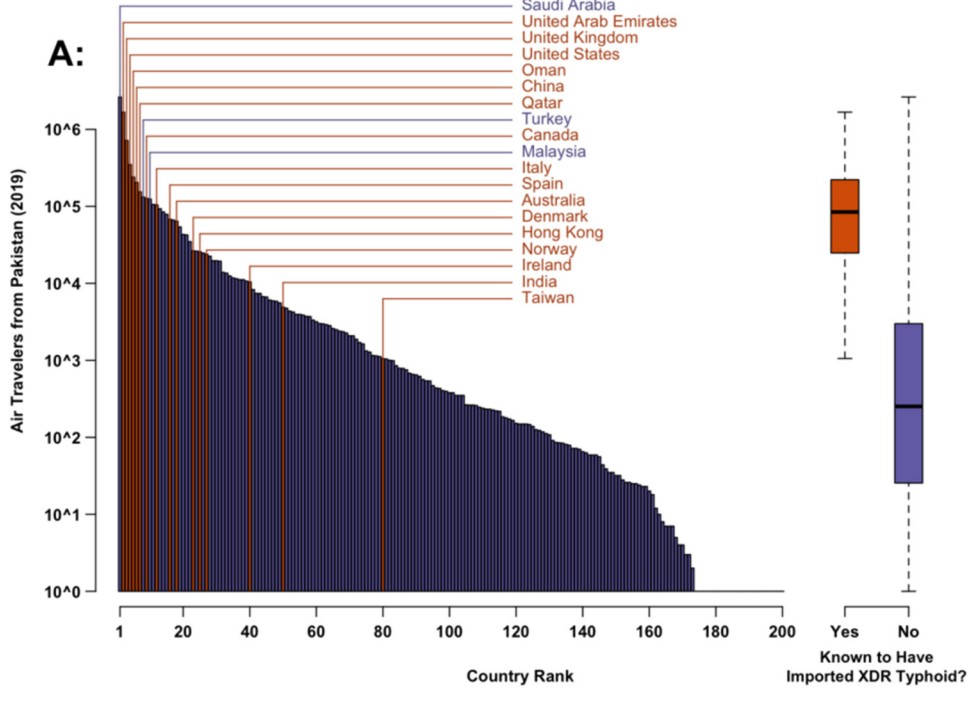

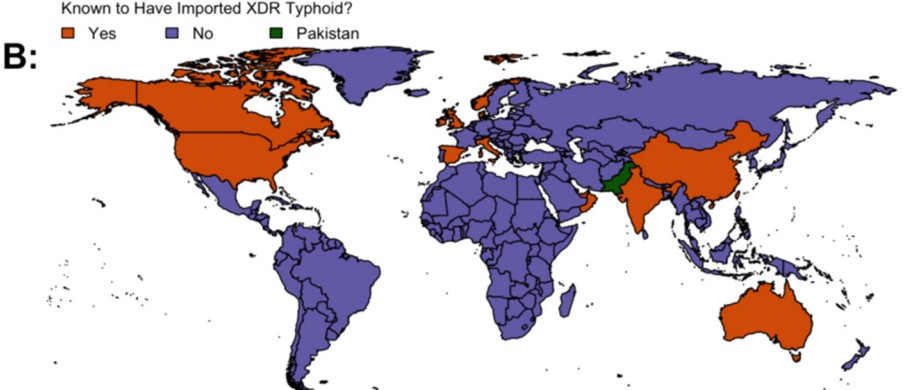

**Fig. 1 | Air travel from Pakistan to countries with and without known XDR Salmonella Typhi importations. A** The barplot (left) and boxplot (right) show the volume of 2019 air travel from Pakistan to countries that are (orange, *n* = 16 countries) and are not (purple, *n* = 184 countries) known to have imported extensively drug resistant (XDR) *S*. Typhi cases to date (*n* = 200 countries overall). The number of travelers has been log10-transformed for interpretability. **B** Map of countries by XDR typhoid importation status. Map data from the Natural Earth project. Source data is provided as a Source Data file.

outbreak in Pakistan persists (a risk exacerbated by population displacement and interrupted access to safe drinking water following recent climate change-associated flooding events[45]), and the popularity of international air travel continues to grow, then additional cases of XDR *S*. Typhi will almost certainly be exported globally in the coming years; data on recent air travel patterns can reveal the most likely recipient countries. However, seven of the ten most visited destinations from Pakistan are high-income countries without endemic typhoid transmission, and the introduction of XDR *S*. Typhi does not pose a substantial threat to public health if it is unable to spread locally in the community.

To quantify the efficiency of *S*. Typhi transmission in each country, we examined three sets of estimates of the annual incidence of typhoid fever cases[1,2,46]. To summarize the available evidence and minimize the impact of any model-specific biases, we derived the median estimated incidence per 100,000 person-years (PY) across these studies. Using the median incidence estimate, we identified 57 countries with a high burden of typhoid fever (100 to <500 cases/100,000 PY), and six with a very high burden (≥500 cases/100,000 PY).

For context, these 63 countries received ~340,000 (4.5%) of the 7.7 million international air travelers from Pakistan in 2019 (Fig. 2), with 20 countries receiving more travelers than Taiwan (1057), the least-visited country known to have imported a case of XDR *S*. Typhi (Supplementary Table 1).

Of the ten most popular destination countries from Pakistan with a high or very high burden of typhoid fever (Fig. 4 and Supplementary Table 3), four countries are in South Asia (Afghanistan, Bangladesh, India, and Sri Lanka), four are in Southeast Asia (Indonesia, Malaysia, the Philippines, and Thailand), and two are in Eastern Africa (Uganda and Kenya). Of these countries, typhoid fever incidence was very high in Afghanistan and Bangladesh (median estimates of 744.3 and 545.1 cases per 100,000 PY, respectively), while Malaysia and Thailand (high typhoid burden countries, with 128.6 and 199.5 cases per 100,000 PY, respectively) received the most travelers from Pakistan in 2019 (124,384 and 93,396, respectively). Together, Malaysia and Thailand alone accounted for 63.6% of air travel from Pakistan to other high and very high typhoid burden countries in 2019 combined. By virtue of having both efficient local typhoid transmission and a relatively high

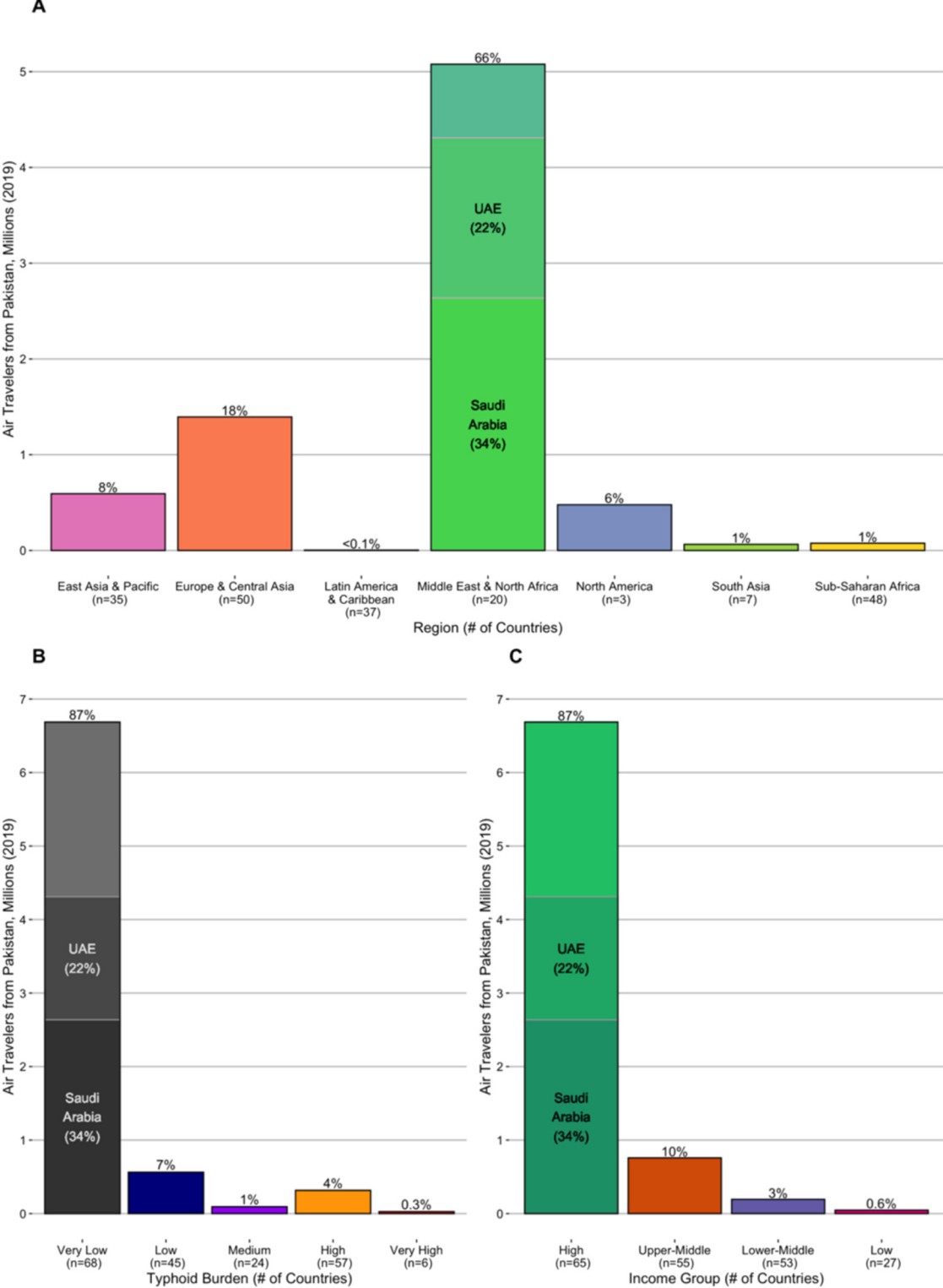

**Fig. 2 | International air travel from Pakistan by typhoid burden, income group, and region.** Bar plots show the volume and percent of international air travel from Pakistan to destination countries in 2019 grouped by (**A**) region, (**B**) typhoid burden, and (**C**) income classification. Saudi Arabia and the United Arab Emirates (UAE), which together received more than 50% of international air travel from Pakistan, are highlighted. Source data is provided as a Source Data file.

volume of travel from Pakistan, these countries may be considered at highest risk for future outbreaks of XDR *S*. Typhi.

For some countries, the volume of incoming air travel from Pakistan may not fully capture the risk of XDR *S*. Typhi introduction. Pakistan shares land borders with Afghanistan, China, India, and Iran. Typhoid fever is endemic in each of these countries, and has a high and

very high burden in India and Afghanistan, respectively, under our framework. Air travel volume alone already suggests that Afghanistan and India are more likely to import XDR *S*. Typhi than almost all other countries with a high or very high typhoid burden, and factoring in land travel can only increase our assessment of the importation risk. According to the International Organization for Migration, 3.37 million

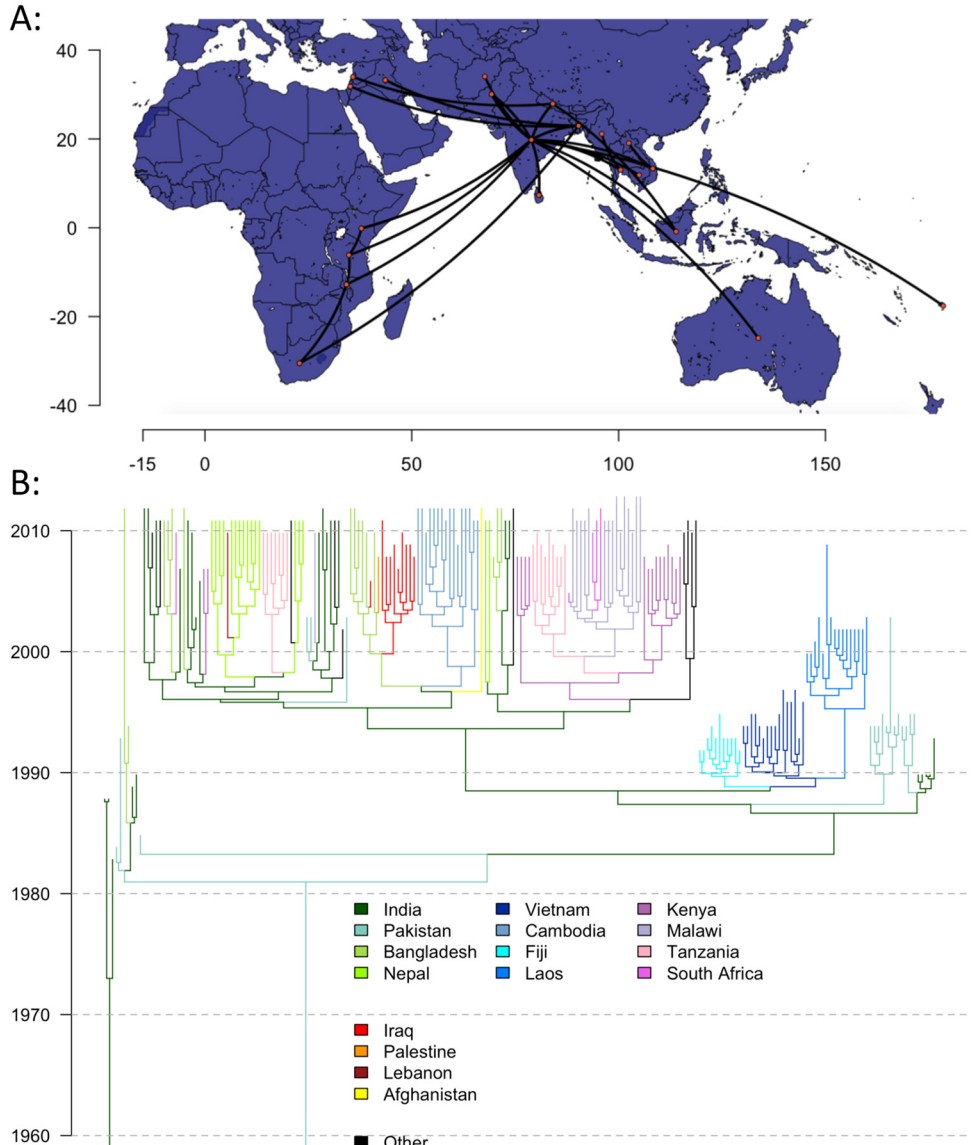

**Fig. 3 | Phylogeography of the H58 haplotype of *Salmonella* Typhi.** A time-scaled discrete phylogeography of the H58 haplotype of *Salmonella* Typhi. The map (**A**) and phylogeny (**B**) correspond to a maximum clade credibility (MCC) tree of 210 H58 genomes sampled over a 30-year period (1983–2013) from 21 countries. Model fitting and inference was performed in BEAST. Map data from OpenStreetMap. Source data is provided as a Source Data file.

trips from Pakistan to Afghanistan were made in 2021 across all entry points[47], which exceeds the number of air travelers from Pakistan to any individual country in 2019 (Supplementary Table 2). We could not identify a specific figure for land travel across the India-Pakistan border, although reports suggest that it is considerably less active than the Afghanistan-Pakistan border, due to longstanding tensions between the two countries[48].

## Discussion

We find that the volume of international air travel is strongly associated with XDR *S.* Typhi importations from Pakistan as well as the phylogeographic estimated movement rate of H58 *S.* Typhi between countries. After validating this association, we cross-referenced the observed rate of incoming air travel from Pakistan against local typhoid fever burden estimates to identify the countries at highest risk of a major outbreak or sustained transmission of XDR *S.* Typhi. This group includes most of South Asia, as well as several countries in Southeast Asia and Sub-Saharan Africa. Afghanistan in particular stands out as a country which not only has one of the highest estimated

typhoid fever burdens, but may also receive the most travelers from Pakistan once trips by air and land are both accounted for, raising major concerns about a typhoid outbreak caused by XDR *S.* Typhi in a country already facing multiple health and humanitarian crises[49]. It may make sense to prioritize investments laboratory surveillance systems and improved water and sanitation infrastructure in these countries, as well as Pakistan itself, to track and prevent the spread of XDR and other *S.* Typhi lineages. Countries should also consider introducing typhoid conjugate vaccines (TCVs), which to date have been introduced in Pakistan, Nepal, Liberia, Zimbabwe, Malawi, Samoa, Fiji's Northern Division, and India's Navi Mumbai municipality[23,50,51].

It is surprising that Saudi Arabia has not reported a single imported case of XDR *S.* Typhi, despite receiving more air travelers from Pakistan in 2019 (2.6 million) than any other country. Similarly, the United Arab Emirates (UAE) received the second most air travelers from Pakistan in this period (1.6 million) but has only documented one case of XDR *S.* Typhi to date[30]. By comparison, the UK[27] and US[32] have each reported over 60 travel-associated cases of XDR *S.* Typhi, despite

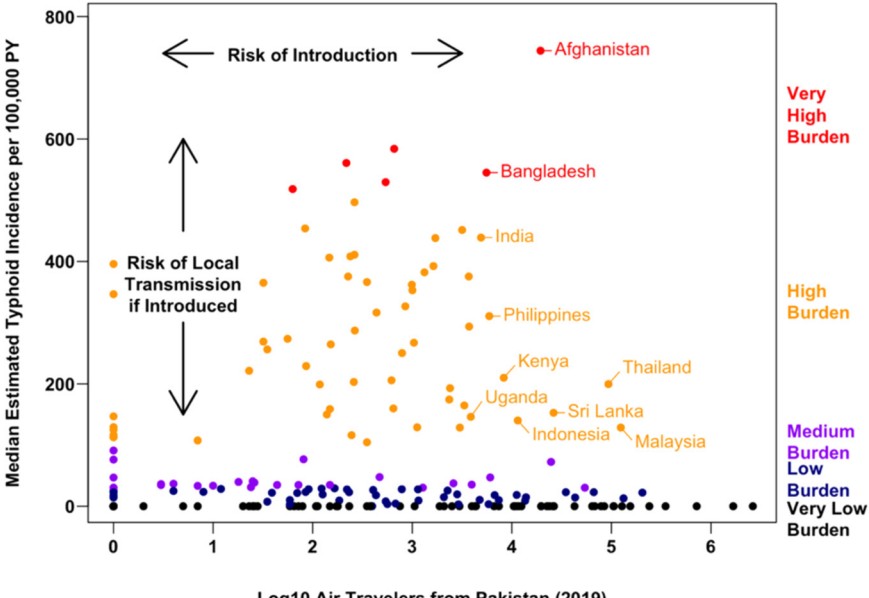

**Fig. 4 | Air travel from Pakistan and estimated typhoid incidence by country.** Each point corresponds to one of 200 countries; countries further to the right received more air travelers from Pakistan in 2019 (a proxy for the risk of XDR typhoid introduction), and countries closer to the top of the graph have a higher estimated typhoid burden (a proxy for the risk of onward transmission and local outbreaks or establishment conditional on XDR typhoid being introduced). PY person-year. Source data is provided as a Source Data file.

receiving less air travel from Pakistan (1.07 million air travelers to both countries in 2019). Even Ireland, which receives less than 1% of Saudi Arabia's and the UAE's annual travel flow from Pakistan, has diagnosed XDR *S.* Typhi in multiple travelers[52]. This suggests that the two most populous Gulf states may be detecting and reporting a relatively small fraction of imported XDR *S.* Typhi cases.

Previous phylogeographic analyses of H58 *S.* Typhi identified South Asia (particularly India) as the initial locus of H58 transmission, which later disseminated to Southeast Asia and Fiji by the early 1990s, East Africa by the late 1990s, and the Middle East by the mid-2000s[12,13]. Outbreaks in East Africa were found to be driven by multiple introductions from South Asia, as well as a localized wave of transmission propagating southward from Kenya through Tanzania, Malawi, and South Africa. We observed the same patterns of geographical spread in our analysis (Fig. 3), and demonstrated a strong association between these patterns and the volume of air travel between countries. This finding supports our use of air travel data to evaluate a country's risk of importing XDR typhoid cases in the future.

While our analysis focuses on the risk of XDR typhoid spreading from Pakistan to other regions, the de novo emergence of XDR in areas with active *S.* Typhi transmission is another potential threat. Prior to 2016, sporadic cases of XDR typhoid were diagnosed in patients from Bangladesh[53] and Iraq[54], although neither of these were linked to outbreaks or endemic transmission, possibly because of an inability to stably maintain the resistance-encoding plasmids. While the identification of hotspots for de novo XDR emergence in the future is outside of the scope of this paper, investments in surveillance, water, sanitation and hygiene improvements, and vaccination should be effective against both imported and locally emergent strains of XDR *S.* Typhi.

Our analysis is subject to a number of limitations. By using the annual volume of air travel from Pakistan as a proxy for the risk of XDR *S.* Typhi introduction to each country, we implicitly assume that the prevalence of XDR *S.* Typhi infection is roughly uniform across travelers to different countries. In practice, the prevalence of infection may vary by destination country if there are systematic differences in travelers' living conditions, vaccination status, timing of travel (due to seasonality in typhoid transmission), or location of residence within

Pakistan. Similarly, our risk assessment does not account for subnational differences in either the volume of incoming travel from Pakistan or the efficiency of onward typhoid transmission at travelers' final destination. While it may be possible to account for these factors in future risk assessments, our results indicate that even simple counts of air travel volume, aggregated by year and origin/destination country, are strongly associated with the global spread of drug-resistant typhoid.

We also assume that the relative volume of air travel from Pakistan to each country will not dramatically differ between 2019 and future years. A retrospective analysis of the IATA travel dataset supports this assumption, as the number of trips from Pakistan in 2019 is highly correlated with that in previous years ($r > 0.9$ for all years 2010–2018, Supplementary Fig. 3). Furthermore, our phylogeographic analysis found that even though the global spread of H58 *S.* Typhi took place over decades following its emergence in India in the 1980s, the estimated volume of air travel in 2010 was strongly associated with this haplotype's rate of movement between countries. The relative popularity of destinations for international air travelers may be primarily driven by factors which do not evolve quickly on yearly timescales, such as geographic proximity, population size, economic activity, and language and cultural ties[43].

Finally, variation in sequencing effort over time and between countries could have biased our phylogenetic analysis. While sampling bias generally cannot be completely eliminated, we attempted to mitigate its impact by independently analyzing multiple subsamples of H58 genomes. We stratified the sampling process by country to balance the distribution of samples and prevent any individual country from being overrepresented. We also mandated the inclusion of several of the oldest available H58 genomes in each subsample, to balance the growing availability of sequence data over time and inform rooting. The subsampling process is described in detail in the "Methods" section. Each phylogeographic analyses inferred a similar relationship between air travel and rate of H58 movement (Supplementary Fig. 2), demonstrating the robustness of these estimates to sampling bias.

XDR *S.* Typhi poses a significant public health threat globally, not just within Pakistan. Our risk assessment may provide a useful

framework to inform the targeting of surveillance activities and typhoid control measures.

## Methods

### Assessing the country-level risk of XDR typhoid outbreaks and establishment

In this analysis, we consider two distinct forms of geographic risk posed by XDR typhoid: the risk of importation from Pakistan, and the risk of efficient local transmission following the introduction of cases.

As a measure of the rate of movement between countries, we analyzed IATA data on international air travel between 2010 and 2019. IATA data accounts for ~90% of global commercial flight passenger itineraries, with the remaining 10% modeled using market intelligence. IATA does not account for passenger volumes or destinations on unscheduled charter flights. We selected all international trips originating in Pakistan, including both direct routes and those with transfers, and aggregated the number of travelers to each final destination country in each year. These counts are based on anonymized records of passenger itineraries derived from global ticket sales on commercial airlines and scheduled charter flights[55]. We used the volume of air travel from Pakistan to each country in 2019 as our main proxy for the risk of XDR typhoid importation. To validate the stability of relative air travel patterns over time, we calculated the correlation between countries' incoming air travel from Pakistan in 2019 and that in each year from 2010 to 2018.

We used national-level estimates of typhoid fever incidence as a proxy for the local transmissibility of *S.* Typhi. To represent the current state of knowledge and minimize the impact of model-specific biases, we took the median estimate across three previously published analyses: Antillón et al.[1], the 2017 Global Burden of Disease (GBD) study[2], and Kim et al.[46].

### Phylogeographic identification of predictors of historical H58 lineage movement

To validate our use of the volume of air travel from Pakistan as a proxy for the risk of importing XDR typhoid, we used Bayesian phylogeographic methods[42,56] to test the relationship between air travel volume and the inter-country transition rate of H58 *S.* Typhi in the decades following its emergence in India[12,13].

For this analysis, we used 1804 *S.* Typhi consensus genomes (849 from the H58 haplotype) previously analyzed by Wong et al.[13]. We augmented this dataset, which only contains H58 isolates from Vietnam ($n = 183$), Fiji ($n = 11$), and Nepal ($n = 1$) before the year 2000, by including 83 early H58 sequences (sampled between 1983 and 1995) described in Carey et al.[57]. Both groups took several steps to validate and control the quality of their sequencing data: in each analysis, raw paired-end Illumina reads were mapped onto a CT18 reference genome[58], and candidate SNP's were only retained if they had a quality score >30 and a coverage depth ≥4 or 5 (for Wong et al. and Carey et al., respectively). A breakdown of the sequences in our combined dataset by country and H58 status is given in Supplementary Data 1. First, we aligned these sequences against the *S.* Typhi CT18[58] reference genome using snippy[59] (v4.0.2). We then used gubbins[60] (v2.3.4) to identify the full set of non-recombinant variable sites and fit a maximum-likelihood (ML) phylogeny with RAxML v8[61], specifying a generalized time-reversible (GTR) evolutionary model with gamma-distributed site heterogeneity. The resulting filtered alignment included 2980 unique sites across the 932 H58 sequences. We extracted the subtree of H58 sequences in R (v3.6.3) using the ape[62] (v5.6-1) and TreeTools[63,64] (v1.8.0) R packages. We evaluated temporal signal in the ML phylogeny by fitting linear regression models to the sampling year and root-to-tip distance of our sequences (Supplementary Fig. 4). The strength of the temporal signal was relatively weak across all isolates ($r = 0.092$), but much stronger for the H58 sequences ($r = 0.602$), which is consistent with previous analyses[13].

To reduce geographic bias and computational complexity, we generated five subsamples of H58 genomes. Each subsample was generated by randomly selecting 16 genomes from each of the 21 countries with H58 isolates in our dataset. If a country did not have more than 16 H58 genomes to sample from, we included all available genomes in each subset. To ensure proper rooting, with H58 first emerging in South Asia in the 1980s[12,13,57], we added the 11 earliest H58 sequences (those sampled through 1990: nine from India, and two from Pakistan) to all subsamples (if they had not already been randomly selected for inclusion), for a total of 209–210 sequences per subsample covering 1251–1304 variable sites.

Finally, we performed three independent analyses on each of the five H58 genome subsamples in BEAST[56] (v1.10.4) and pooled the resulting estimates together. Sequences were dated according to the year of sampling (subannual collection dates were not consistently available), and the country of sampling was specified as a discrete trait. Each analysis assumed a GTR model of nucleotide evolution with estimated base frequencies and gamma-distributed site heterogeneity, an uncorrelated lognormal relaxed clock, and a Bayesian skyline demographic model for the change in population size over time. The RAxML-generated maximum-likelihood tree was used to define the initial tree topology. To model the geographic spread of H58 typhoid, we specified each genome's country of sampling as a discrete trait, and parameterize[42] the transition rate in terms of a GLM with a log link, such that:

$$\log(y_{o,d}) = \sum_{j=1}^{J} (I_j \beta_j + (1 - I_j)\mu_j) x_{j,o,d} \tag{1}$$

where:

$y_{o,d}$ is the transition rate from origin country $o$ to destination country $d$; $J$ is the total number of candidate predictors proposed in the model; $I_j$ is an indicator variable denoting whether predictor $j$ is included in the model ($I_j = 1$) or not ($I_j = 0$). For each predictor $j$, the prior probability of inclusion is $P(I_j = 1) = 1 - 0.5^{1/J}$, such that the prior probability of including no predictors in the model is 0.5; $\beta_j$ is a fitted coefficient representing the effect of predictor $j$ on the transition rate $y$ conditional on predictor $j$'s inclusion in the model ($I_j = 1$). The prior distribution of $\beta_j$ is normal with mean 0 and standard deviation 2; $\mu_j$ is a randomly sampled value from the prior distribution of $\beta_j$ (and therefore has an expected value of 0), and acts as the coefficient for predictor $j$ when it is not selected for inclusion in the model ($I_j = 0$); $x_{j,o,d}$ is the value of predictor $j$ for origin country $o$ and destination country $d$.

We propose $J = 2$ candidate predictors: a binary variable (taking on values of 0 or 1) indicating the presence or absence of a land border between the origin and destination countries, and the annual volume of air travel from country $o$ to country $d$.

We were unable to obtain IATA air travel data between all 21 countries with H58 isolates in our dataset, so we used modeled estimates of air travel volume in 2010 from Huang et al.[43]. These estimates are based on a number of factors, including the distance and number of necessary transfers between origin and destination airports, and the population and economic activity in the vicinity of each. To match the geographic specificity of our sequences, we mapped each airport to a country and aggregated the volume of travel between them. The modeled volume of outgoing air travel from India to each country was strongly correlated with the volume observed by IATA in 2010 ($r = 0.83$). Presumably, our estimate of the effect of air travel volume on the rate of H58 typhoid movement will be biased toward the null (no effect), due to any inaccuracies in the modeled volume of air travel between countries, and variation in the relative volume of air travel over time (we assume that travel volume in 2010 is imperfectly correlated with travel over the entire period H58 *S.* Typhi has circulated). For ease of interpretation, we log-transformed and standardized air travel volume. Full details on the GLM methodology can be found in Lemey et al.[42].

Analyses were performed in BEAST via Markov chain Monte Carlo (MCMC), with classic priors and operators, chains of 200 million iterations, a 20 million iteration burn-in period, and sampling of parameters every 5000 iterations. Finally, we used LogCombiner (v1.10.4) to pool samples from the three independent analyses for each of the five sets of genomes, and evaluated the posterior distribution, fit, and convergence of model parameters in Tracer[56] (v1.10.4). For each subsample of genomes, all parameters had an effective sample size >200.

## Statistics and reproducibility
We did not perform any formal experiments or interventions in this study, only analyses of preexisting data. No data were excluded from the analyses. No statistical method was used to predetermine the sample size in these analyses. Investigators were not blinded to any outcomes while performing data analyses.

## Reporting summary
Further information on research design is available in the Nature Portfolio Reporting Summary linked to this article.

## Data availability
The full genome sequences used in our phylogeographic analysis and corresponding metadata have been deposited in the pathogen.watch database at https://pathogen.watch/collection/vlnn0nzwjfr0-walker-et-al-2023. All other non-genomic data needed to replicate this analysis can be accessed at github.com/joewalker127/XDR_Typhoid. Source data are provided with this paper.

## Code availability
All code needed to replicate this analysis can be accessed at https://github.com/joewalker127/XDR_Typhoid.

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

## Acknowledgements

This work was supported by a grant from the Bill and Melinda Gates Foundation (INV-030857, to J.W. and V.E.P.). All other authors received no specific funding for this work. We would also like to acknowledge the map data copyrighted OpenStreetMap contributors for using the map data available from https://www.openstreetmap.org.

## Author contributions

The study was jointly conceived by J.W. and V.E.P. J.W. conducted the analysis and wrote the paper under the supervision of V.E.P. C.C., N.D.G., M.C., and S.B. provided input on phylogenetic modeling methods and the selection of genomic data. K.K. and I.I.B. provided the air travel data and assisted in its interpretation. All authors reviewed, edited, and approved the manuscript.

## Competing interests

K.K. is the founder of BlueDot, a social enterprise that develops digital technologies for public health. All other authors declare no competing interests.
