## [Peer Review File · Nature Communications]

REVIEWER COMMENTS

Reviewer #1 (Remarks to the Author):

This is a neat paper on an important issue. I enjoyed reading it and believe it should be published. Though, I have a few queries and recommendation to the authors, listed in no particular order below.

I appreciate the authors might have tried to stay away from the fundamental phylogeographic aspects, maybe to avoid an overlap with the paper by Wong et al. 2015, but this hurts the readability of the article. At the risk of creating some redundancy with earlier work, the phylogenetics aspects have to be covered at least fleetingly., and properly described.

Salmonella enterica has long been considered to be a clonal organism. This is actually not true and all serovars undergo homologous recombination (at least within lineages). As such, weeding out (or at least testing for) recombining regions in the genome (using e.g., a 'first principle' search for homoplasies, Clonalframe or Gubbins) seems warranted prior to the phylogenetic modelling. This may also increase the temporal signal, which was weak in Wong et al.

The Beast analysis behind the time-tree (fig 3) needs to be explained and detailed. As is, this is an unreproducible black box. At the very least, evidence needs to be provided for temporal signal (possibly post-recombination cleaning) and for convergence of the runs.

It may also be interesting to comment on the estimated dates associated to key nodes in the tree. A history of the sequential acquisition of drug-resistance by the H58 haplotype might be a neat addition to the paper, unless it simply recapitulated previous findings.

I chose not to quibble about whether underlying sampling biases may colour the results, as I doubt they call the main conclusions in question. Sampling biases are inherent to such analysis, and it is often largely pointless to try to control for them. This being said, a paragraph in the discussion about sampling biases seems warranted.

One question I have, and that other readers may wonder about too, is why did XDR Typhi apparently evolve only once. A priori, conditions may be suitable for multiple independent emergences of XDR Typhi in various settings. It may be interesting for the authors to speculate on this in the discussion.

I strongly suggest moving the three tables to the supplementary material.

Reviewer #2 (Remarks to the Author):

This paper provides important evidence of possible transmission of XDR S. Typhi (cause of Typhoid fever) through Airtraffic originating from Pakistan, a key endemic setting. Whilst the modeling and assumptions are justified, it will be important to note that many factors contribute to disease and AMR genes dissemination within countries and between countries and regions. There are a number of concerns that require to be addressed.

ABSTRACT

1. Please address the issue of nomenclature of Salmonella enterica serotype Typhi, throughout the paper: Typhi rather than typhi, is the expectation for serotype.
2. We require a sentence on outcome/result of this study and public health implication as part of the conclusion.

RESULTS

Air Travel Patterns Predict the International Spread of Drug-Resistant S. typhi

Comment: What is the difference in data presented in Table 1 and 2? I see none, and one should be deleted, and information consolidated into a single database. This is also repeated for Tables 3, with several datapoints recurring. Authors should endeavor to consolidate their findings into a flowing narrative backed by Tables/Figures.

DISCUSSION

1. The discussion lacks a crucial consideration for the phylogenetic and phylogeographic analyses that should provide unequivocal data and deductions on temporal and spatial relatedness of H58 strains from originating country (Pakistan) and the several destination (recipient) countries.

a). Frequency of transfers of specific subtypes and haplotypes from Pakistan (origin) and destination countries

These elements strengthen the hypothesis for origin and transfer of H58 XDR haplotype

b). Spatio-temporal dynamics of these transfers/transmission in relation to phylogenetic analyses from origin to destination countries

2. Line sentence starting: It may make sense to prioritize investments in laboratory surveillance systems and improved water and sanitation infrastructure in these countries, as well as Pakistan itself, to track and inhibit the spread of XDR and other *S. typhi* lineages.

Comment: Consider replacing inhibit with Reduce/curtail/minimize/prevent the spread of XDR

Reviewer #3 (Remarks to the Author):

Walker et al., performed a study to assess the global risk of typhoid outbreaks caused by XDR *S. typhi*.

The study impact is high and the work has a significance in the field. But a few issues should be addressed:

-Did the authors perform the genomes assembly using snippy? So they downloaded the raw reads? Please include more detail in the methods.

- The author performed a whole tree with the different haplotypes to then extract only a subtree of H58 haplotype. I suggest including this tree as supplemental material. All the 849 sequences clustered together in a monophyletic clade? Please include the number of sequences and the countries used in the different analyses.

- What demographic model and molecular clock model was selected?

-Did the SNP ascertainment bias was used?

-Why an effective sample size > 100 for the parameters was used? It is recommended to have them > 200.

-All the MCC trees of 5 subsets of genomes were consistent and equal? what the authors meant by this: "Across samples from 15 pooled model replicates..." (line 149)? What are these 15 pooled?

-I think Figure 2 and Tables 1 and 2 are a little bit redundant. I did not understand the need to have all of these elements.

-Since it was chosen to assess only the Pakistan impact as origin, probably it is not a global risk but rather a local risk.

Reviewer #1 (Remarks to the Author):

This is a neat paper on an important issue. I enjoyed reading it and believe it should be published. Though, I have a few queries and recommendation to the authors, listed in no particular order below.

I appreciate the authors might have tried to stay away from the fundamental phylogeographic aspects, maybe to avoid an overlap with the paper by Wong et al. 2015, but this hurts the readability of the article. At the risk of creating some redundancy with earlier work, the phylogenetics aspects have to be covered at least fleetingly., and properly described.

We have revised our paper to provide more details on the methods and results of the phylogeographic analysis. Our specific changes are described below.

Salmonella enterica has long been considered to be a clonal organism. This is actually not true and all serovars undergo homologous recombination (at least within lineages). As such, weeding out (or at least testing for) recombining regions in the genome (using e.g., a 'first principle' search for homoplasies, Clonalframe or Gubbins) seems warranted prior to the phylogenetic modelling. This may also increase the temporal signal, which was weak in Wong et al.

Thank you for this suggestion. We have added a new step to our phylogenetic analysis: after using *snippy* to align our sequences against the CT18 reference genome, we use *gubbins* to simultaneously 1) extract an alignment of non-recombinant variable sites, and 2) fit a maximum likelihood tree. After extracting the H58 sequences and subtree, we used these two outputs as the sequence alignment and the starting tree topology for our BEAST analyses. We have revised the methods section of the manuscript accordingly.

The Beast analysis behind the time-tree (fig 3) needs to be explained and detailed. As is, this is an unreproducible black box. At the very least, evidence needs to be provided for temporal signal (possibly post-recombination cleaning) and for convergence of the runs.

To evaluate temporal signal across all 1,887 *S. Typhi* sequences in our maximum likelihood phylogeny, and among the 932 H58 sequences used in our BEAST analysis, we fit linear regression models against sequences' date of sampling and their divergence from the root in our maximum likelihood tree. Our findings were very similar to those of *Wong et al.*, who detected a moderate temporal signal in the H58 sequences, but no meaningful signal across the full *S. Typhi* tree. This is not surprising, since the H58 sequences represent clonal expansion and epidemic spread over a relatively short period of time. Furthermore, our detection of moderate temporal signal among the H58 sequences supports our decision to proceed with using them in a time-scaled BEAST analysis to investigate the predictors of phylogeographic spread. We

have added a new Supplementary Figure 4 depicting the results of our temporal analyses, which we discuss in a revised 5th paragraph of the methods section.

It may also be interesting to comment on the estimated dates associated to key nodes in the tree. A history of the sequential acquisition of drug-resistance by the H58 haplotype might be a neat addition to the paper, unless it simply recapitulated previous findings.

In this paper, we examine the emergence of the H58 haplotype because it is a useful system for testing the association between air travel volume and the rate of *S. Typhi* lineage movement from country to country. Validating this relationship for H58 supports our use of air travel data (in combination with data on local typhoid transmission) to identify the countries at greatest risk of XDR typhoid outbreaks in the future. While the timeline of drug resistance in the H58 haplotype is interesting to consider, we believe that it is beyond the scope of our study. Fortunately, other analyses have investigated the timing of resistance acquisition within the H58 lineage, notably *Carey et al.* (DOI: 10.1101/2022.10.03.510628). We have added a discussion paragraph in which we comment on the timing of the appearance of the H58 haplotype in different geographic regions, and the consistency of our results with previous analyses.

I chose not to quibble about whether underlying sampling biases may colour the results, as I doubt they call the main conclusions in question. Sampling biases are inherent to such analysis, and it is often largely pointless to try to control for them. This being said, a paragraph in the discussion about sampling biases seems warranted.

In most situations, it isn't possible to fully eliminate sampling biases from phylogenetic analyses. Of course, if sampling bias is present in the underlying studies which produced the genomic data we used, this bias will be inherited by our analysis. To mitigate the impact of sampling bias along geographic lines, we generated 5 subsamples of H58 genomes, each containing a maximum of 16 randomly selected genomes from each country, in an effort to balance the distribution of samples and prevent any individual country from being overrepresented. Phylogeographic analyses performed on these five subsamples generated similar results (see the newly added Supplementary Figure 2), demonstrating the robustness of our findings to geographic sampling bias. We have added a brief paragraph on sampling biases to the discussion.

One question I have, and that other readers may wonder about too, is why did XDR *Typhi* apparently evolve only once. A priori, conditions may be suitable for multiple independent emergences of XDR *Typhi* in various settings. It may be interesting for the authors to speculate on this in the discussion.

Thank you for this suggestion: we have added a paragraph on this topic to the discussion section.

I strongly suggest moving the three tables to the supplementary material.

We have modified Figure 1 such that it now lists the 10 most popular destinations for air travel from Pakistan, in addition to the 15 countries known to have imported XDR typhoid from Pakistan (7 overlapping countries). This information was previously given in tables 1 and 2, which we have moved to the supplement. We have also moved table 3 to the supplement, as this information (the 10 high typhoid burden countries that receive the most air travel from Pakistan) is also given in Figure 4.

Reviewer #2 (Remarks to the Author):

This paper provides important evidence of possible transmission of XDR S. Typhi (cause of Typhoid fever) through Airtraffic originating from Pakistan, a key endemic setting. Whilst the modeling and assumptions are justified, it will be important to note that many factors contribute to disease and AMR genes dissemination within countries and between countries and regions. There are a number of concerns that require to be addressed.

ABSTRACT

1. Please address the issue of nomenclature of Salmonella enterica serotype Typhi, throughout the paper: Typhi rather than typhi, is the expectation for serotype.

We have made this change throughout the text.

2. We require a sentence on outcome/result of this study and public health implication as part of the conclusion.

Our abstract now concludes with the following: "Future outbreaks of XDR typhoid are most likely to occur in countries which can support efficient local S. Typhi transmission and have strong travel links to regions with ongoing XDR typhoid outbreaks (currently Pakistan). Public health activities to track and mitigate the spread of XDR S. Typhi should be prioritized in these countries."

RESULTS

Air Travel Patterns Predict the International Spread of Drug-Resistant S. typhi

Comment: What is the difference in data presented in Table 1 and 2? I see none, and one should be deleted, and information consolidated into a single database. This is also repeated for Tables 3, with several datapoints recurring. Authors should endeavor to consolidate their findings into a flowing narrative backed by Tables/Figures.

Table 1 lists the 15 countries which are known to have imported XDR typhoid in the past. Meanwhile, Table 2 lists the 10 countries which received the most air travel from Pakistan in 2019. While there is substantial overlap between the two tables (7 countries are present in both), consistent with our hypothesis that air travel volume from Pakistan is a key risk factor for the importation of XDR typhoid, they are not identical. Some countries in Table 1 have imported XDR typhoid cases despite receiving relatively few air travelers from Pakistan (Norway, Ireland, India, Taiwan, etc), while some countries in Table 2 have not reported XDR typhoid cases despite being popular travel destinations from Pakistan (Saudi Arabia, Turkey, and Malaysia). To simplify our results and improve readability, we have moved all 3 tables to the supplement, and now indicate Saudi Arabia, Turkey, and Malaysia (the only top 10 countries for air travel from Pakistan to not have not reported an XDR typhoid importation) in Figure 1 and the results section.

DISCUSSION

1. The discussion lacks a crucial consideration for the phylogenetic and phylogeographic analyses that should provide unequivocal data and deductions on temporal and spatial relatedness of H58 strains from originating country (Pakistan) and the several destination (recipient) countries.

a). Frequency of transfers of specific subtypes and haplotypes from Pakistan (origin) and destination countries

These elements strengthen the hypothesis for origin and transfer of H58 XDR haplotype

b). Spatio-temporal dynamics of these transfers/transmission in relation to phylogenetic analyses from origin to destination countries

Previous phylogeographic analyses of H58 *S. typhi*, such as those performed by Wong et al and da Silva et al, show that this haplotype emerged in South Asia in the 1980's before spreading worldwide, first to Southeast Asia, then later to East Africa, and most recently to the Middle East. We observed the same patterns of spread in our phylogeographic analysis, and also showed for the first time that H58 movement patterns were associated with the volume of air travel between countries. Confirming this association for H58 *S. typhi* helps to validate our use of air travel data to identify the countries which are most likely to import XDR typhoid in the future. We have added a brief discussion paragraph on the H58 movement patterns we observed in the phylogeographic analysis, their consistency with previous studies, and what they indicate about the role of air travel in typhoid lineage movement and future XDR outbreaks.

2. Line sentence starting: It may make sense to prioritize investments in laboratory surveillance systems and improved water and sanitation infrastructure in these countries, as well as Pakistan itself, to track and inhibit the spread of XDR and other *S.*

typhi lineages.

Comment: Consider replacing inhibit with Reduce/curtail/minimize/prevent the spread of XDR

Thank you for this suggestion. We have replaced “inhibit” with “prevent” here.

Reviewer #3 (Remarks to the Author):

Walker et al., performed a study to assess the global risk of typhoid outbreaks caused by XDR *S. typhi*.

The study impact is high and the work has a significance in the field. But a few issues should be addressed:

-Did the authors perform the genomes assembly using snippy? So they downloaded the raw reads? Please include more detail in the methods.

We did not assemble the genomes from raw reads. We downloaded consensus genomes (which *Wong et al.* and *Carey et al.* generated from raw reads) from public repositories, and used *snippy* to align these contigs against the full CT18 reference genome. We have revised the 5th paragraph of our methods section to clarify this.

- The author performed a whole tree with the different haplotypes to then extract only a subtree of H58 haplotype. I suggest including this tree as supplemental material. All the 849 sequences clustered together in a monophyletic clade? Please include the number of sequences and the countries used in the different analyses.

We have added the maximum likelihood tree as a new Supplementary Figure 5. The H58 sequences (colored red) cluster together, away from the other genotypes. We have also attached a new file “Supplementary_Data_1.csv” listing the number of H58, non-H58, and total sequences from each country used in our phylogenetic analysis. We direct readers to this file in the 5th paragraph of the methods section.

- What demographic model and molecular clock model was selected?

In the 7th paragraph in the methods section, we state that we used a Bayesian skyline demographic model and an uncorrelated relaxed clock with a lognormal prior.

-Did the SNP ascertainment bias was used?

We use consensus *S. Typhi* genomes from *Wong et al.* and *Carey et al.* for the phylogenetic component of our analysis. Each of these groups took several steps to validate and control the quality of their sequencing data: in each analysis, raw paired-end Illumina reads were mapped onto a CT18 reference genome, and candidate SNP's were only retained if they had a phred quality score > 30 (indicating >99.9% probability of a correct base call) and a coverage depth of at least 4 or 5 (for *Wong et al.* and *Carey et al.*, respectively). We have added this information on sequence data quality to our methods section.

-Why an effective sample size > 100 for the parameters was used? It is recommended to have them > 200.

Across the 5 genome subsamples, ESS > 200 was achieved for the vast majority of parameters. We were able to achieve ESS > 200 for all parameters by increasing the chain length and sampling frequency. We have revised the text to clarify this.

-All the MCC trees of 5 subsets of genomes were consistent and equal? what the authors meant by this: "Across samples from 15 pooled model replicates..." (line 149)? What are these 15 pooled?

The "15 pooled model replicates" represent 3 independent BEAST runs for each of the 5 subsets of H58 genomes. We have revised the 2nd and 3rd paragraphs of the results section to clarify our meaning and confirm that posterior estimates of the association between air travel volume and lineage movement rates were comparable between model replicates (as shown in the newly added Supplementary Figure 2).

-I think Figure 2 and Tables 1 and 2 are a little bit redundant. I did not understand the need to have all of these elements.

To simplify our results and improve readability, we have moved all of our tables to the supplement (see above).

-Since it was chosen to assess only the Pakistan impact as origin, probably it is not a global risk but rather a local risk.

While the current *burden* of XDR typhoid has been largely concentrated in Pakistan, there are many countries in the world which presumably could support outbreaks of their own (as evidenced by their high non-XDR typhoid incidence), and many of these countries also have relatively strong connections with Pakistan via air travel (Table 2). Given the potential for outbreaks in typhoid-endemic areas outside of Pakistan, we consider XDR typhoid to be a global health risk.

REVIEWERS' COMMENTS

Reviewer #1 (Remarks to the Author):

The authors ave satisfactorily addressed all my concerns and suggestions. I'd be happy for the manuscript to be published as is.